# Novel model to predict risk of invasive fungal infection and fungal prophylaxis timing

Xinyu Zuo,[1] Xinyuan Ma,[2] Miao Zhang,[1] Richeng Mao,[3] Jiexian Ma[1]

**ABSTRACT** Patients on immunosuppressive drugs or those who are critically ill are at high risk for invasive fungal infections (IFIs). The assessment of IFI risk and the initiation of prophylaxis in these patients remain unclear. A nomogram model was developed to evaluate clinical and immune indicators in relation to IFI risk and validated by immuno-compromised patients. High-risk patients, as identified by the model, were selected for a prospective randomized study to assess the efficacy of the model and timing of fungal prophylaxis initiation. Patients deemed high risk received either fluconazole or a placebo until IFI occurrence or risk downgrade for 90 days. We compared the incidence of IFI and mortality between the two groups. The nomogram, created from a training cohort ($n = 384$), included age, IgG level, and CD4$^+$ cell count as predictive indicators of IFI and was validated in a separate cohort ($n = 281$) with an area under the curve of 0.723. A total of 265 patients were recruited into the prospective study, with 163 high-risk patients randomly assigned to receive either fluconazole ($n = 83$) or a placebo ($n = 80$). The model had a positive predictive value of 48.8% and a negative predictive value of 90.2%. High-risk IFI defined by this model could be reduced to the low-risk cohort level with fluconazole prophylaxis ($P < 0.01$). This nomogram model reliably predicts the risk of IFI and timing of mycoprophylaxis in immunocompromised patients. Targeted mycoprophylaxis significantly reduces the incidence of IFI, particularly yeast infections, and may help to prevent fungal colonization.

**IMPORTANCE** We still lack enough evidence to decide when and how to begin fungal prophylaxis in immunocompromised patients. A model based on immune function indicator is an effective tool for predicting risk of invasive fungal infection (IFI) in immunocompromised patients. Patients were collected to evaluate the performance of the model retrospectively and prospectively.

**CLINICAL TRIALS** This study is registered with the Chinese Clinical Trial Registry as ChiCTR2400079810.

**KEYWORDS** nomogram, invasive fungal infection, prophylaxis, scoring system, immunocompromised patients

Invasive fungal infections (IFIs) are severe systemic infections resulting from the colonization of yeasts or molds in deep-seated tissues. Unlike superficial fungal infections, IFIs are life-threatening and are associated with high morbidity and mortality rates (1). Recent statistics indicate that approximately 1.9 million patients experience an acute IFI annually, while an estimated 3 million people globally suffer from chronic severe fungal infections. These infections are often life-threatening, with more than 1.6 million deaths per year attributed to all fungal diseases (2). Invasive candidiasis accounts for nearly 70% of all IFIs worldwide, followed by cryptococcosis (20%) and aspergillosis (10%) (2, 3).

**Peer Reviewer** Shalini Upadhyay, Medanta The Medicity, Gurgaon, Haryana, India

Address correspondence to Jiexian Ma, majiexian@fudan.edu.cn, or Richeng Mao, njxiaomao@163.com.

Xinyu Zuo and Xinyuan Ma contributed equally to this article. Xinyu Zuo collected patient data, analyzed the data, and wrote the paper; Xinyuan Ma collected and analyzed the data.

The authors declare no conflict of interest.

Populations at the highest risk for contracting opportunistic fungal infections include organ transplant recipients, patients with hematologic malignancies undergoing stem cell transplantation, individuals with acquired immunodeficiency syndrome, diabetics, burn patients, neoplastic disease patients, and patients receiving long-term immunosuppressive therapy (4). The development of targeted therapies has expanded the range of drugs that can cause varying levels of immunosuppression, such as chemotherapy agents, glucocorticoids, calcineurin inhibitors, tumor necrosis factor-α blockers, lymphocyte-specific monoclonal antibodies, Bruton's tyrosine kinase inhibitors, phosphoinositide 3-kinase inhibitors, and new therapies like Chimeric Antigen Receptor T-cell therapy (5).

To reduce the incidence and mortality of IFIs, antifungal prophylaxis is widely recommended. It is essential to identify the target population for antifungal prophylaxis, as inappropriate or unnecessary prophylaxis can lead to side effects and economic waste. Additionally, selective pressure from antifungal prophylaxis, along with advances in molecular testing, may contribute to the emergence of less common fungal pathogens, including rare yeasts and molds that are often resistant to currently available classes of antifungal treatments (6, 7).

Previous studies have indicated that patients with prolonged neutropenia (≤500/µL for ≥7 days), such as those with acute myeloid leukemia or myelodysplastic syndrome undergoing remission-induction chemotherapy, or patients with severe aplastic anemia, remain at the highest risk for developing invasive fungal diseases (8, 9). In non-neutropenic patients, the risk of IFIs is closely linked to host immune status and the type of fungal species (10, 11). Most pathogenic fungi are opportunistic pathogens that cause disease under immunosuppressed conditions such as human immunodeficiency virus infection, cancer, chemotherapy, transplantation, and the use of immunosuppressive drugs. Current guidelines emphasize the importance of initiating antifungal prophylaxis in immunocompromised patients (12). However, there is still a lack of gold standards to predict the incidence of IFI and to guide fungal prophylaxis. The exact indicators or appropriate tools to evaluate immunocompromised status and assess the risk of IFI remain undefined.

Our previous study investigated immune-related risk factors for IFI, identifying immunoglobulin G levels and NK cell counts as potential indicators (13). However, due to the limited sample size of that cohort, we were unable to establish a predictive model for IFI. To address this limitation, we have now enrolled nearly 700 hospitalized infected patients in Huashan and Huadong Hospitals in China. This larger cohort will enable us to develop a clinical risk model to predict the incidence of IFI, focusing on the roles of humoral and cell-mediated immunity.

## MATERIALS AND METHODS

### Study patient population

#### Study patients for nomogram modeling

We retrospectively reviewed the medical records of 1,000 patients (aged ≥18 years) from the Department of Infectious Diseases at Huashan Hospital and the Department of Hematology at Huadong Hospital in Shanghai, China, between 1 June 2022 and 30 June 2023. Patients included in the retrospective study met the following criteria: they had received chemotherapy or immunosuppressive drugs in the past 3 months, or had chronic conditions such as diabetes, cancer, organ transplantation, or chronic infections. Assessments of humoral and cellular immunity, including immunoglobulin levels, peripheral lymphocyte counts, and flow cytometry analysis, were conducted for all patients. Patients with neutropenia, a history of invasive fungal infection, or those who had received antifungal prophylaxis were excluded from the study. The collected clinical data comprised age, gender, disease diagnosis, type of transplant, chemotherapy regimen, chronic diseases, and other clinical complications.

### Study patients for prospective clinical trials

Patients for the prospective clinical trials were selected from the immunocompromised patient group consistent with the criteria for nomogram modeling from July 2023 to July 2024. Exclusion criteria included neutropenia (absolute neutrophil count of 500 cells per cubic millimeter or less), history of invasive fungal infection, receipt of antifungal prophylaxis, clinically significant hepatic or renal dysfunction (two times above the normal level), abnormal QT interval corrected for heart rate (QTc interval), a baseline Eastern Cooperative Oncology Group performance status score of more than 2 (in bed more than half of the day), a history of hypersensitivity or idiosyncratic reactions to azoles, or a requirement for medications with potential adverse interactions with azoles. Both the retrospective review and the prospective clinical trials were approved by the Institutional Ethical Committee of Huadong and Huashan Hospitals (Ethical Committee Number: 2022K095; Approved: March 2022) in accordance with the Helsinki Declaration of 1975, revised in 2008. All patients provided informed consent in accordance with the Declaration of Helsinki. The prospective clinical trials were registered online: www.chictr.org.cn (ChiCTR2400079810).

### Immune function monitoring

According to the guidelines (14, 15), after recovering from neutropenia, only patients suffering from grade III to IV graft-versus-host disease in allogeneic hematopoietic stem cell transplantation should receive antifungal prophylaxis. In this study cohort, none of the cases required implementation of antifungal chemoprophylaxis according to current clinical practice guidelines. To evaluate humoral immune function, we measured IgG, IgM, and IgA levels using a turbidimetric immunoassay. Cellular immune function was assessed through the proportions of $CD4^+/CD8^+$, $CD4^+$, $CD8^+$, $CD19^+$, and NK cells, measured by flow cytometry (BD Corporation). A total of 50,000 cells were counted each time. Cell counts were calculated using a peripheral blood test, and the proportions were measured by flow cytometry.

### Clinical evaluation and definition of IFI

According to the revised definitions by the European Organization for Research and Treatment of Cancer (EORTC)/the National Institute of Allergy and Infectious Diseases Mycoses Study Group (MSG) Consensus Group (11, 15), a probable IFI necessitates the presence of both clinical and mycological criteria. In this study, patients presenting with clinical criteria of IFI (e.g., typical signs on lung computed tomography, images showing sinonasal or central nervous system infection) required supplementary evidence from mycological criteria. This evidence included (i) direct microscopic examination employing both conventional smear microscopy and fungal immunofluorescence staining. In the direct smear analysis, specimens were evaluated for fungal elements such as spores, hyphae, and other characteristic structures. Fluorescent staining using Calcofluor white or Blankophor to enhance detection; (ii) microbiological confirmation via fungal culture on Sabouraud dextrose agar and Brain Heart Infusion agar, incubated at 30°C for up to 4 weeks; (iii) real-time polymerase chain reaction targeting the 18S rRNA gene, using the primers F: 5′-GAT AAC GAA CGA GAC CTC GG-3′ and R: 5′-AGA CCT GTT ATT GCC GCG C-3′ (Primer Premier 5.0), with DNA extraction performed using the QIAamp DNA extraction kit (Qiagen, Germany) according to the manufacturer's protocol; (4) an indirect assay for galactomannan (GM) antigen in plasma, serum, bronchoalveolar lavage fluid, or cerebrospinal fluid; (v) next-generation sequencing (NGS) for the detection of fungal elements in blood, bronchoalveolar lavage fluid (BALF), cerebrospinal fluid, or urine.

### Feature selection, model building, and evaluation

The predictive models for invasive fungal infection were constructed as follows: First, patients were randomly divided into training and validation groups at a ratio of 4:3. Univariate Cox regression analyses and logistic regression algorithms were employed to

identify clinical features significantly associated with invasive fungal infection. Statistically significant variables in the univariate analysis were included in the logistic model and refined through a stepwise elimination process. The signature was calculated using a multivariable logistic regression model by linearly combining the most predictive features, weighted by their respective coefficients. A nomogram was built by proportionally converting regression coefficients of each predictor in the combined model to a 0- to 100-point scale. The receiver operating characteristic (ROC) curve and area under ROC curve (AUC) were used to evaluate the predictive accuracy of the established models in the training and validation cohorts. Decision curve analysis (DCA) was conducted to estimate the clinical utility of the IFI signature, clinical risk factors, and nomogram by quantifying net benefits at different threshold probabilities in the validation data set. The decision curves of the treat-all and treat-none strategies served as references in the DCA. The clinical usefulness and benefits of the nomogram were estimated by DCA plots. Furthermore, based on the risk score and cutoff points on the ROC curve, all patients were stratified into low- and high-risk groups. The workflow of this study is illustrated in Fig. 1.

## Prospective study design

In this prospective, randomized trial, patients undergoing chemotherapy, receiving immunosuppressive drugs, or suffering from chronic diseases were recruited and underwent humoral and cellular immune measurements for risk stratification based on the nomogram model. High-risk IFI patients were enrolled in this study and randomly assigned by a random table method, in a 1:1 ratio, to receive either fluconazole for the prevention of invasive fungal infections or a placebo. Patients with liver or renal dysfunction, prior IFI, or an inability to tolerate the oral study drug were excluded from the study. All patients included in this study underwent humoral and cellular immune measurements monthly. Fluconazole prophylaxis was discontinued if an invasive fungal infection occurred or if the risk stratification for IFI was decreased to low risk. Patients began antifungal treatment if IFI occurred. Patients were followed for 90 days after randomization. An independent data review committee of infectious disease experts, who were blinded to the treatment assignments, reviewed and classified all cases of fungal infection as proven or probable according to the consensus criteria of the EORTC and the MSG (11).

## Administration of the study drug

Study patients received 400 mg of fluconazole (Diflucan, Pfizer) or a placebo orally once daily. Patients in either group were permitted to receive amphotericin B or another systemic agent as empirical antifungal therapy for suspected IFI. All patients underwent comprehensive evaluations for the presence of IFI at the beginning and end of prophylaxis. During the treatment phase—defined as the period from randomization to 7 days after the last dose of the study drug was administered—if a patient exhibited any sign or symptom of infection, including fever, a complete clinical and mycologic evaluation was performed. Surveillance blood specimens were collected once weekly for the assessment of liver and renal function. *In vitro* susceptibility testing of fungal isolates was conducted by staff at a central laboratory in Huashan Hospital. If necessary, NGS detection was performed by an authoritative tripartite testing agency, with results validated by experts in infectious diseases.

The primary efficacy endpoint was the incidence of proven or probable invasive fungal infection during the treatment phase, as adjudicated by an expert panel blinded to the treatment assignments, according to consensus criteria from the EORTC and MSG (11). Secondary endpoints included treatment failure within 90 days after randomization. Treatment failure was defined as meeting any of the following criteria: (i) proven or probable invasive fungal infection or death possibly or probably related to fungal infection or any other reason; (ii) the occurrence of an adverse event possibly or probably related to the study treatment resulting in the discontinuation of treatment; and (iii)

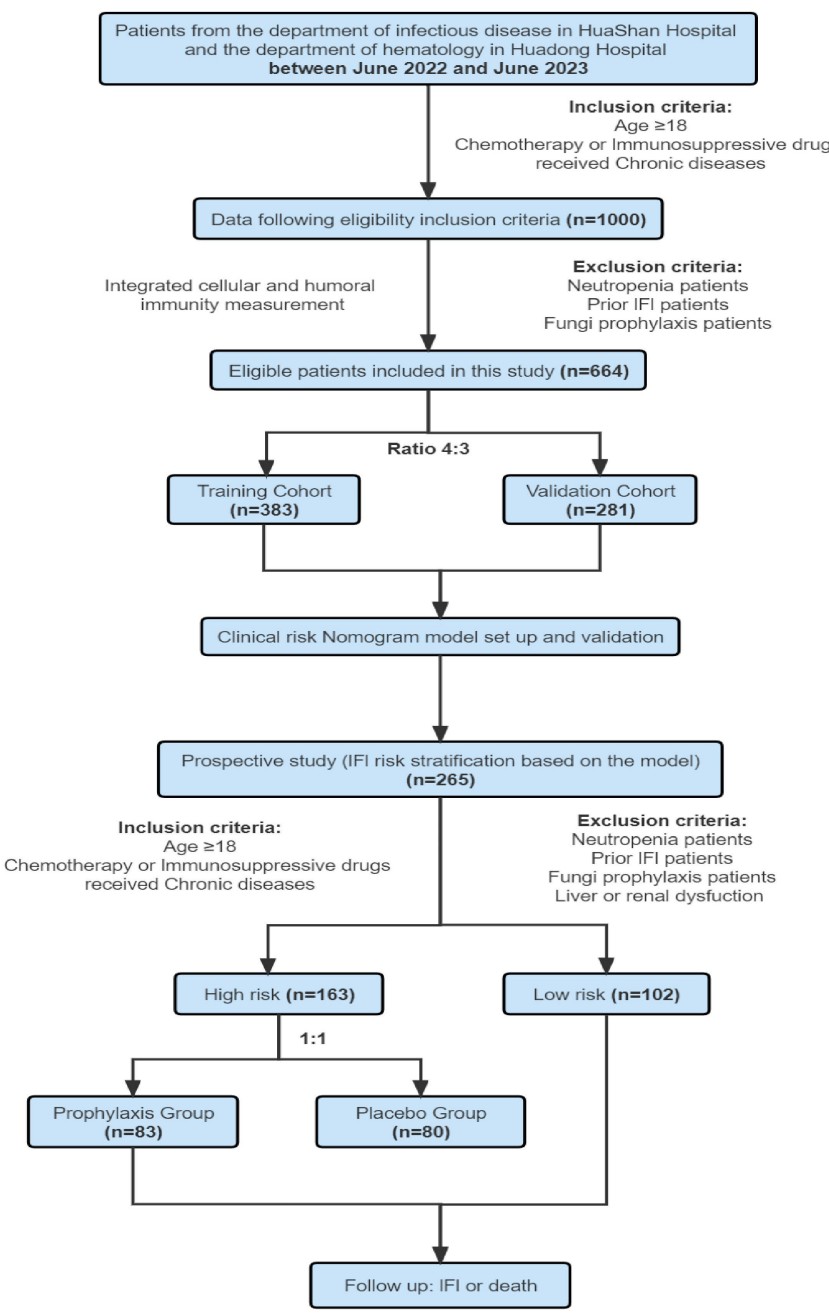

**FIG 1** The flow of the whole study.

withdrawal from the study with no additional follow-up. Withdrawal from the study due to adverse events or loss to follow-up was excluded from the infection incidence analysis and survival analysis, but still included in the clinical treatment failure analysis. Survival was evaluated 90 days after randomization, with analyses conducted for overall survival, time to death from any cause, time to death related to fungal infection, and survival without proven or probable IFI. Time to IFI and time to first use of empirical antifungal therapy were also assessed.

## Statistical analysis

For nomogram modeling, ROC curve, AUC, and Harrell's concordance index (C-index) were used to assess model discrimination, while the calibration plot graphically

**TABLE 1** Clinical characteristics in patients with nomogram modeling

| Characteristic | All patients ($n = 664$) | Training cohort ($n = 383$) | Validation cohort ($n = 281$) | P |
|---|---|---|---|---|
| Age in years, mean (range) | 58.71 (18–90) | 58.61 (18–90) | 58.86 (18–88) | 0.857 |
| Male sex, n (%) | 458 (69) | 263 (68.7) | 195 (69.4) | 0.865 |
| Central venous catheter, n (%) | 238 (35.8) | 135 (35.2) | 103 (36.7) | 0.743 |
| Chemotherapy, n (%) | 99 (14.9) | 66 (17.2) | 33 (11.7) | 0.060 |
| Transplantation, n (%) | 16 (2.4) | 10 (2.6) | 6 (2.1) | 0.801 |
| Diabetes, n (%) | 135 (20.3) | 81 (21.1) | 54 (19.2) | 0.560 |
| Chronic pulmonary diseases, n (%) | 99 (14.9) | 55 (14.4) | 44 (15.7) | 0.660 |
| Cancer, n (%) | 142 (21.4) | 89 (23.2) | 53 (18.9) | 0.174 |
| WBCs, mean (range) | 8.55 (0.70–70.8) | 8.56 (0.70–70.80) | 8.53 (1.41–27.38) | 0.957 |
| Neutrophils, mean (range) | 6.76 (0.16–67.2) | 6.67 (0.16–67.2) | 6.89 (0.51–67.2) | 0.633 |
| Lymphocytes, mean (range) | 1.42 (0.08–41.32) | 1.58 (0.08–41.32) | 1.2 (0.10–8.20) | 0.078 |

evaluated the calibration of the nomogram in both training and validation cohorts. The C-index ranges from 0.5 to 1.0, with 0.5 indicating random chance and 1.0 demonstrating perfect discrimination. All analyses were conducted using R software (version 3.6.3), and $P$ values less than 0.05 were considered statistically significant.

In the prospective study, the primary efficacy analysis was based on an intention-to-treat basis, using data from all patients who underwent randomization. Continuous variables were expressed as medians and interquartile ranges, while categorical variables were presented as frequencies and percentages (%). The Kaplan-Meier method was used to evaluate time to death from any cause, time to death related to fungal infection, time to proven or probable fungal infection, time to first use of empirical antifungal therapy, and survival free from proven or probable invasive fungal infection. The survival benefit was assessed using the chi-square and log-rank tests. The number of patients needed to be treated to prevent one fungal infection and one death (numbers needed to treat) was calculated using STATA software (version 18.0) (16).

## RESULTS

### Clinical characteristics and prognostic factors of IFI in immunocompromised patients

A total of 665 patients from the Infectious Diseases Department of Huashan Hospital who met the inclusion criteria were selected for this study. This cohort included 201 individuals diagnosed with IFI and 464 individuals without IFI. These patients were randomly divided into a training cohort ($n = 384$) and a validation cohort ($n = 281$) at a 4:3 ratio. Demographics and baseline health conditions showed no significant differences between the two groups (Table 1).

Subsequently, we compared variables such as age, gender, disease type, treatment, comorbidities, and laboratory parameters, including levels of IgG, IgA, and IgM, counts of T cells, CD4$^+$ T cells, CD8$^+$ T cells, B cells, NK cells, and white blood cell count (WBC), between the IFI and non-IFI groups in the training cohort to explore potential risk factors for invasive fungal infection. Univariate analysis identified age, IgG level below 6.5 g/L, CD4$^+$ T-cell count, CD8$^+$ T-cell count, and pulmonary diseases as significantly associated with the occurrence of IFI ($P < 0.05$, Table 2). Multivariate logistic regression analysis further validated independent risk factors for IFI, with age, IgG level below 6.5 g/L, and CD4$^+$ T-cell count retaining significance ($P < 0.05$, Table 2). An overview of all factors and their relationship to IFI is provided in Table 2. Finally, a Cox regression analysis was performed, including only the three significant predictors, to demonstrate the fitted coefficients and hazard ratios of each predictor in the model.

## Nomogram construction for prediction risk of IFI and validation of the predictive model

The three independent prognostic factors (age, IgG level, and CD4+ T-cell count) were incorporated into a predictive model to estimate the probability of invasive fungal infection (Fig. 2). This model was visualized using a nomogram, with an illustrative example of a hypothetical patient based on age (below 40 years old, 40-59 years old, 60–80 years old, above 80 years old), IgG level below 6.5 g/L, and CD4+ T-cell count (below 200/µL, 200–500/µL, above 500/µL). Points assigned for age categories were 0, 28, 79, and 100, respectively. An IgG level below 6.5 g/L was assigned 62 points, and CD4+ T-cell counts below 200/µL, 200–500/µL, and above 500/µL were assigned 70, 65, and 0 points, respectively. For example, a patient aged 70, with an IgG level of 5.5 g/L and a

**TABLE 2** Univariate and multivariate logistic regression analyses for invasive fungal infection patients in the training cohort[a]

| Variables | | Univariate analysis OR (95% CI) | P | Multivariate analysis OR (95% CI) | P |
|---|---|---|---|---|---|
| Age | <40 | 1 | | 1 | |
| | 40–59 | 1.07 (0.34–1.71) | 0.765 | 1.16 (0.49–1.51) | 0.322 |
| | 60–80 | 2.47 (1.22–4.98) | 0.012 | 2.19 (1.07–4.51) | 0.033 |
| | >80 | 3.92 (1.55–9.92) | 0.004 | 2.78 (1.06–7.32) | 0.039 |
| Sex | Female | 1 | | – | – |
| | Male | 1.03 (0.64–1.66) | 0.910 | – | – |
| Central venous catheter | No | 1 | | – | – |
| | Yes | 1.20 (0.76–1.89) | 0.446 | – | – |
| Diabetes | No | 1 | | – | – |
| | Yes | 1.32 (0.78–2.23) | 0.302 | – | – |
| Chronic pulmonary diseases | No | 1 | | 1 | |
| | Yes | 2.90 (1.64–5.12) | <0.001 | 1.93 (0.98–3.78) | 0.056 |
| Cancer | No | 1 | | – | – |
| | Yes | 1.09 (0.66–1.79) | 0.742 | – | – |
| Chemotherapy | No | 1 | | – | – |
| | Yes | 1.66 (0.95–2.9) | 0.073 | – | – |
| Transplantation | No | 1 | | – | – |
| | Yes | 1.68 (0.47–6.07) | 0.429 | – | – |
| IgG | ≥6.5 | 1 | | 1 | |
| | <6.5 | 3.31 (1.75–6.24) | <0.001 | 2.53 (1.28–5.02) | 0.008 |
| IgA | ≥0.7 | 1 | | – | – |
| | <0.7 | 1.76 (0.73–4.23) | 0.211 | – | – |
| IgM | ≥0.3 | 1 | | – | – |
| | <0.3 | 0.32 (0.07–1.46) | 0.143 | – | – |
| CD4 | >500 | 1 | | 1 | |
| | 200–500 | 2.47 (1.34–4.53) | 0.004 | 2.7 (1.43–5.09) | 0.002 |
| | <200 | 3.75 (2.03–6.91) | <0.001 | 2.95 (1.55–5.61) | <0.001 |
| CD8 | ≥300 | 1 | | 1 | |
| | <300 | 1.67 (1.04–2.70) | 0.035 | 0.77 (0.38–1.55) | 0.455 |
| CD4/CD8 | ≥1 | 1 | | – | – |
| | <1 | 1.57 (0.98–2.51) | 0.063 | – | – |
| B-cell count | ≥1 × 10^9 | 1 | | – | – |
| | <1 × 10^9 | 1.31 (0.84–2.07) | 0.236 | – | – |
| NK cell count | ≥100 | 1 | | – | – |
| | <100 | 1.38 (0.88–2.15) | 0.162 | – | – |

[a]– indicates that the variable was not included in the multivariate analysis.

CD4$^+$ T-cell count of 150/µL, would have a total score of 211 points, corresponding to an approximate 0.62 probability of invasive fungal infection (Fig. 2).

## Clinical validation and application using the predictive model in immuno-compromised patients

### *Validation of the predictive model using retrospective clinical data*

We retrospectively reviewed the clinical data of 281 patients to evaluate the performance of the model. We scored each patient according to the model, then analyzed the infected patients. The performance of this nomogram was assessed using the C-index, AUC, and calibration plots. The C-index of the predictive model was 0.72 in the training cohort and 0.73 in the validation cohort. ROC curves for IFI prediction in both cohorts indicated AUC values of 0.718 for the training cohort and 0.723 for the validation cohort, demonstrating high model discrimination (Fig. 3). Calibration plots also showed excellent agreement between predicted and observed probabilities of IFI, indicating good calibration of the model (Fig. S1).

### *Validation of the model using prospective clinical data*

The study was conducted from July 2023 to July 2024 at two medical centers: Hua-shan and Huadong Hospital in China. A total of 265 patients were screened using the nomogram model to identify high-risk IFI patients. According to the nomogram model and the ROC curve, patients with a total score above 115 points were classified as high-risk IFI patients. Consequently, 163 patients were enrolled as high-risk patients in the prospective study, while 102 patients were categorized as low-risk and their

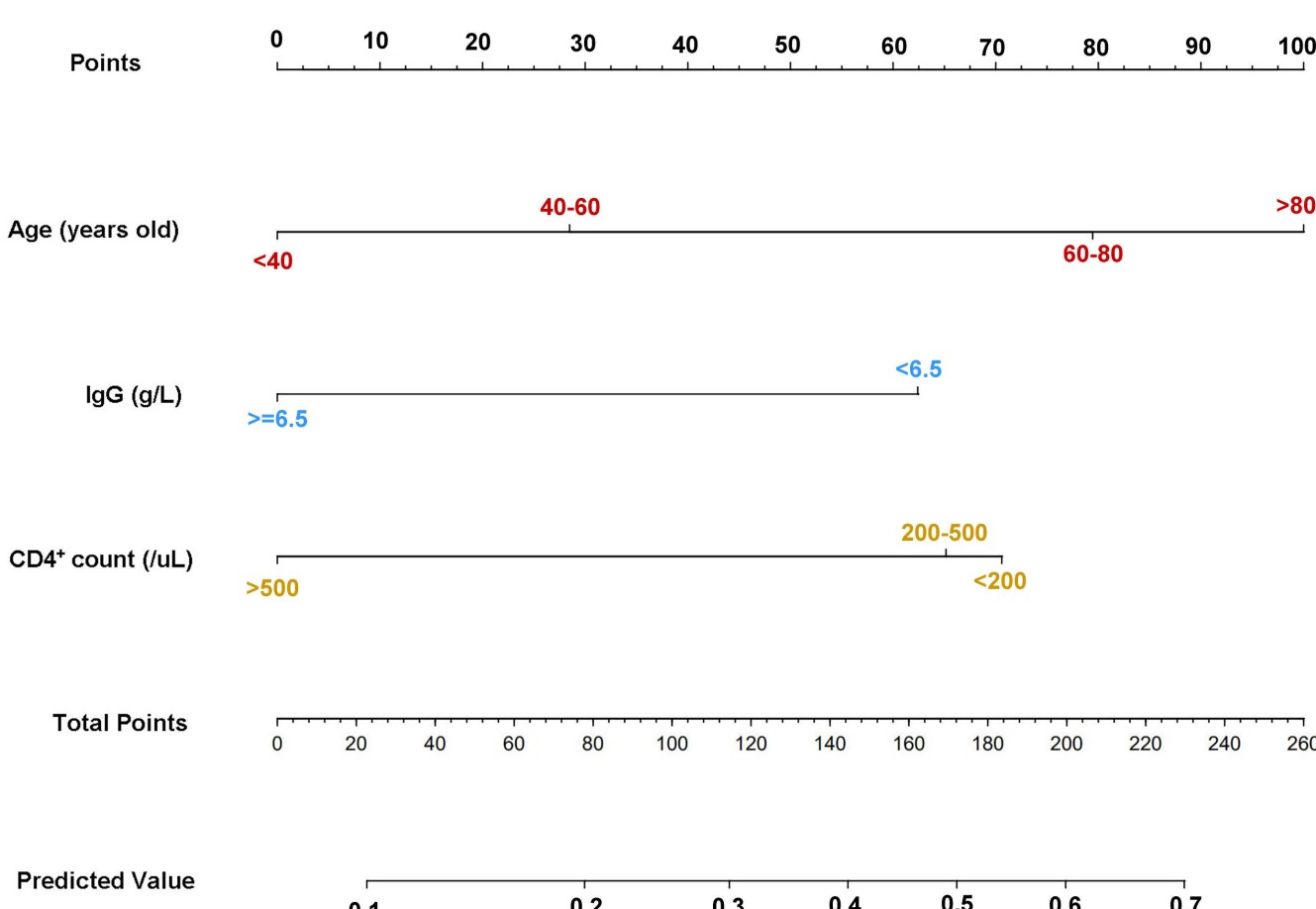

**FIG 2** The final nomogram consisting of age, IgG level, and CD4$^+$ T-cell count is displayed.

subsequent incidence of invasive fungal infection was recorded (Fig. 1). The 163 high-risk patients were randomly assigned to treatment groups: 80 patients received a placebo and 83 patients received fluconazole. All groups exhibited similar characteristics (Table 3). The antifungal treatment regimens for patients with suspected or proven IFI can be found in Table S1. There were no differences in antifungal drug use between the groups. Five patients exited the study due to adverse effects or were lost to follow-up and were excluded from the survival analysis. Thus, each group was included in the final analysis.

During the 90-day period following randomization, any signs of infection, such as fever, in patients receiving either fluconazole or placebo triggered a comprehensive check-up, including C-reactive protein, procalcitonin, G test, GM test, chest CT scan, sputum culture, blood culture, and BALF NGS testing to identify pathogens. Proven IFI was defined by positive sputum culture, blood culture, or pathogen identification via NGS. Probable IFI was indicated by positive G or GM tests and was responsive to systemic antifungal treatment. Proven invasive fungal infections occurred in 37 of the 80 patients (46.2%) in the high-risk group, and 10 developed proven IFI during follow-up in the 102 low-risk patients. The model had a positive predictive value of 48.8% and a negative predictive value of 90.2%.

## Incidence of invasive fungal infection was significantly lowered after fungal prophylaxis determined by this model

Rates of clinical success or failure and reasons for clinical failure are detailed in Table 4. Among the 83 patients in the fluconazole group, 14 were defined as clinical failures: 9 (10.8%) due to proven IFI who required empirical antifungal treatment (including four deaths) (Table 4). Four due to adverse effects of long-term fluconazole administration, such as liver dysfunction, and 1 due to a strong desire to discontinue the trial. In the placebo group, 39 of the 80 patients (48.2%) were classified as clinical failures: 37 due to proven IFI, 2 due to probable IFI, and 7 due to death ($P = 0.004$, Table 4).

Among 80 patients in the placebo group, 39 (48.8%) required antifungal treatment for proven or probable invasive fungal infections. This contrasts significantly with the prophylaxis group, where only 9 of 83 patients (10.8%) experienced IFI ($P < 0.001$).

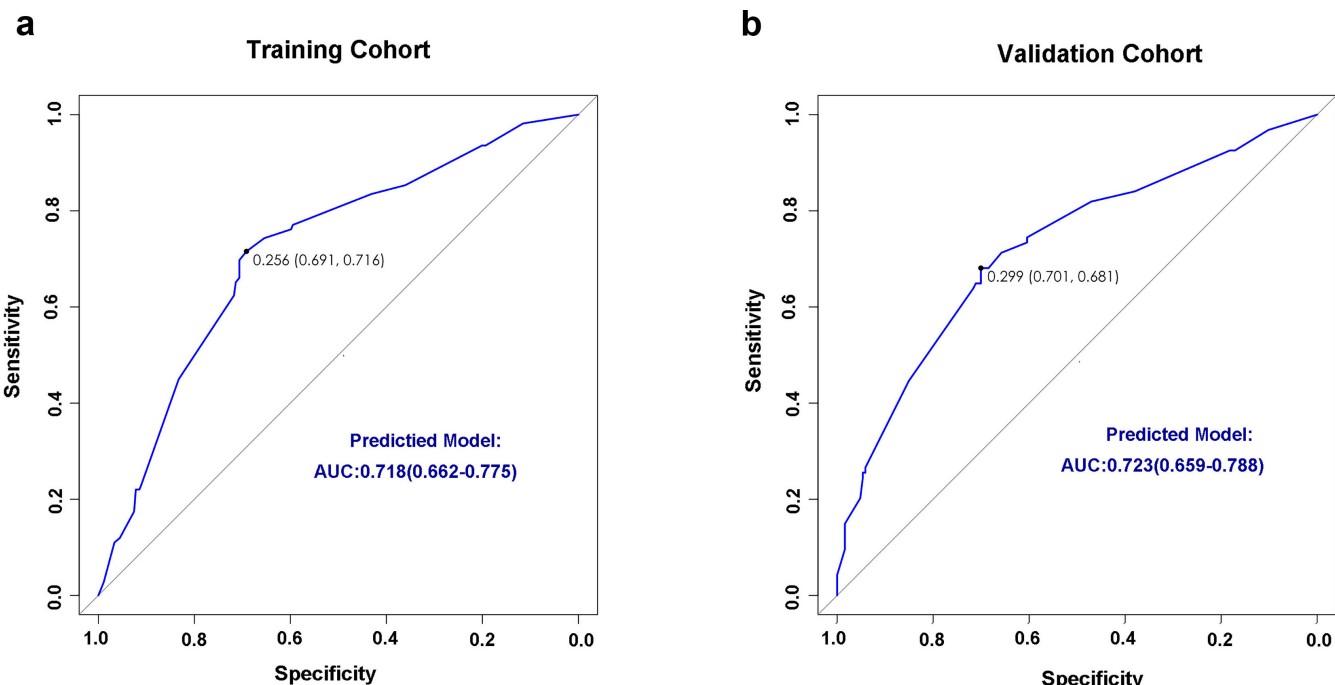

**FIG 3** ROC curves of the IFI risk predictive model in the training (a) and validation (b) dataset. AUC, area under the receiver operator characteristic curve; ROC, receiver operator characteristic.

TABLE 3 Clinical characteristics in high-risk IFI patients with fluconazole prophylaxis and non-prophylaxis

| | High-risk patients (N = 163) | | P value |
|---|---|---|---|
| | Non-fungal prophylaxis (N = 80) | Fungal prophylaxis (N = 83) | |
| Age (mean, range) | 69.40 (31-88) | 66.58 (21-87) | 0.099 |
| Male (yes/total) | 57 (71.3%) | 47 (56.6%) | 0.145 |
| Central venous catheter (yes/total) | 25 (31.3%) | 16 (19.3%) | 0.124 |
| Transplantation (yes/total) | 4 (5%) | 11 (13.3%) | 0.051 |
| Diabetes (yes/total) | 16 (20%) | 16 (19.3%) | 0.936 |
| IgG (mean, range) (gL) | 11.99 (3.13–24.70) | 10.48 (3.46–71.05) | 0.162 |
| CD4 (mean, range) (µl) | 230.33 (15.00–496.00) | 337.24 (20.93–5114.38) | 0.105 |
| WBC (mean, range) ($\times 10^9$L) | 8.18 (1.64–27.38) | 6.59 (1.20–70.80) | 0.208 |
| Neutrophil (mean, range) | 6.67 (0.68–25.97) | 4.55 (0.51–62.05) | 0.059 |
| Lymphocyte (mean, range) | 0.85 (0.12–4.85) | 0.96 (0.17–2.52) | 0.081 |

These findings demonstrate that antifungal prophylaxis effectively reduces IFI incidence, indicating substantial protective benefits for high-risk populations such as immunocompromised patients. The mean (±SD) time to invasive fungal infection was 43 ± 14 days in the placebo group and 39 ± 11 days in the fluconazole group. Kaplan–Meier analysis of the time to invasive fungal infection showed a significant difference in favor of fluconazole ($P < 0.001$) (Fig. 4a). Fluconazole prophylaxis significantly reduced the incidence of IFI to levels comparable to those in the low-risk group ($P < 0.001$, Fig. 4a). Kaplan–Meier analysis of the time to death did not show a significant difference between the placebo and fluconazole groups ($P = 0.34$, Fig. 4b). Table 5 lists the causative pathogens of invasive fungal infections, indicating significantly fewer cases of *Candida albicans* associated with fluconazole prophylaxis compared to placebo ($P < 0.001$).

## DISCUSSION

IFIs are a leading cause of mortality in patients with severe infections. Despite the consensus on the necessity of fungal prevention in immunocompromised patients, there is no standardized definition of immunosuppression, nor are there established monitoring and prevention indicators. The most significant contribution of our study is its focus on patients undergoing immunosuppressive therapy, who may experience varying degrees of immunosuppression. Identifying independent prognostic factors in cellular and humoral immunity for IFIs is crucial for guiding clinical practice.

We developed a nomogram model based on these indicators to score and stratify patients, facilitating targeted prevention for those at high risk, and then validated this model in retrospective and prospective clinical patients. Our prospective study demonstrated that fluconazole prophylaxis is safe, with only a small proportion of patients (4.8%) experiencing reversible liver damage upon discontinuation. The prophylactic use of fluconazole in high-risk patients determined by this model

TABLE 4 Clinical response and reasons for failure during the treatment phase

| Clinical response | Non-fungal prophylaxis patients (placebo) (N = 80) | Fungal prophylaxis patients (N = 83) | P value |
|---|---|---|---|
| Clinical success | 41 | 69 | |
| Clinical failure | 39 | 14 | <0.001 |
| Proven or probable invasive fungal infection | 39 | 9 | <0.001 |
| Adverse event possibly or probably related to study treatment, resulting in discontinuation | 0 | 4 | 0.09 |
| Withdrawal for any reason and loss to follow-up | 0 | 1 | 0.32 |

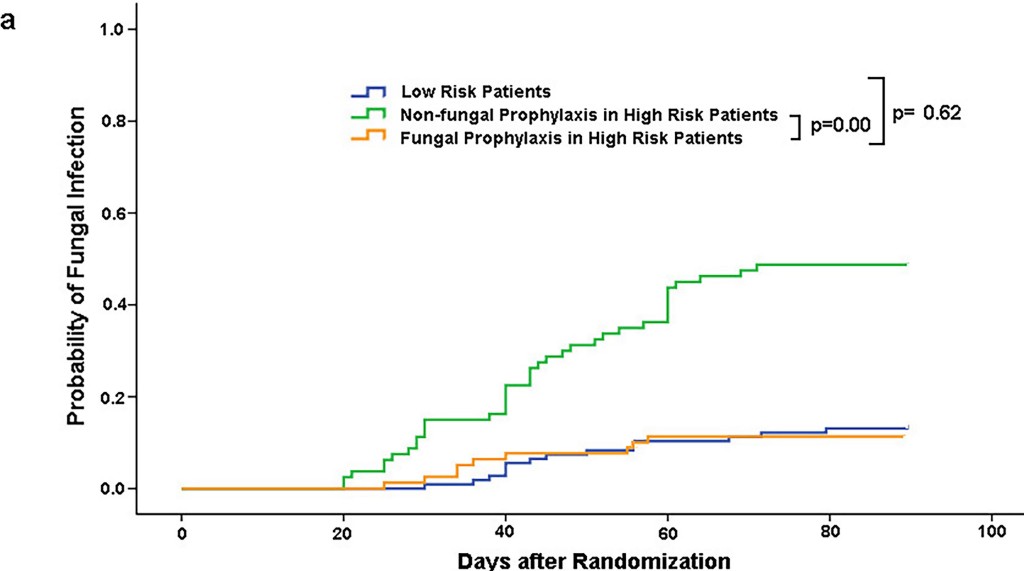

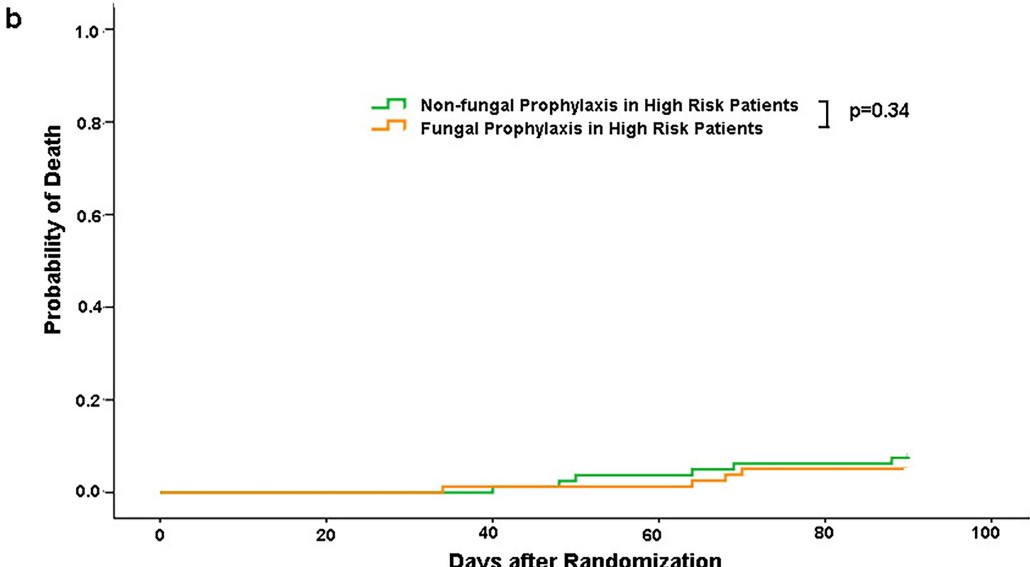

**FIG 4** Kaplan–Meier curves for time to invasive fungal infection (a), death from any cause (b) over the 90 day period after randomization (*P* values were estimated with the log-rank test. Data were censored on the last date of contact or on day 90 after randomization).

significantly reduced the incidence of fungal infections, particularly *Candida albicans*, in high-risk patients, aligning their fungal incidence rate with that of low-risk patients. Although fluconazole prophylaxis did not reduce mortality rates, lowering the incidence of fungal infections in immunocompromised patients remains clinically meaningful. Fungal infections can easily lead to colonization, and high-risk infections may still occur in the future.

In considering the appropriate prophylactic agent, we initially debated whether to use fluconazole or posaconazole. (17)Multiple clinical studies have demonstrated the superiority of posaconazole over fluconazole in reducing infection and mortality rates

**TABLE 5** Proven or probable invasive fungal infection during the treatment phase

| Species | Low-risk patients (N = 102) | High-risk patients (N = 163) | | |
|---|---|---|---|---|
| | | Non-fungal prophylaxis (N = 80） | Fungal prophylaxis (N = 83） | P value |
| Mold | 7 | 8 | 5 | P = 0.40 |
| Aspergillus fumigatus | 5 | 5 | 3 | |
| A. flavus | 0 | 2 | 1 | |
| A. terreus | 0 | 1 | 1 | |
| A. niger | 1 | 0 | 0 | |
| A. tubingensis | 1 | 0 | 0 | |
| Yeast | 5 | 26 | 3 | P < 0.01 |
| Candida albicans | 3 | 12 | 1 | |
| C. glabrata | 0 | 1 | 0 | |
| C. parapsilosis | 0 | 5 | 1 | |
| C. tropicalis | 2 | 6 | 1 | |
| C. krusei | 0 | 2 | 0 | |
| Other | 0 | 5 | 1 | P = 0.1 |
| Pneumocystis jirovecii | 0 | 4 | 1 | |
| Scedosporium | 0 | 1 | 0 | |

in patients with long-term granulocytopenia. However, large-scale clinical studies on fluconazole prophylaxis in immunosuppressed patients are lacking. Considering the cost of the drug and the expenses associated with clinical research, we opted for fluconazole for this study.

Our findings indicated that some patients still experienced breakthrough infections, predominantly caused by *Aspergillus*, *Candida tropicalis,* and *Candida parapsilosis*. These patients often scored higher in risk stratification based on the nomogram model, indicating that they might require broader-spectrum antifungal drugs for prevention. Defining more high-risk patients and selecting the most appropriate antifungal prophylactic agents necessitates larger-scale research, which will be the focus of our future studies.

## ACKNOWLEDGMENTS

This work was supported by the Science and Technology Commission of Shanghai Municipality (grant number 21Y11909000) and Shanghai Health Commission (grant number 202140518), the Elite Project of Huadong Hospital (HD0103). This work was also supported by Shenkang Hospital Development Center (SHDC12025143).

J.M. and R.M. conceived and designed the study. Patient recruitment and clinical care of the patients were performed by X.Z., X.M., R.M., and J.M., X.Z., and X.M. analyzed the data. M.Z. collected the patients' records. The first draft of the manuscript was written by J.M. All authors revised the manuscript and approved the final version.

## AUTHOR AFFILIATIONS

[1]Department of Hematology, Huadong Hospital, Fudan University, Shanghai, China
[2]Shanghai Medical College Fudan University, Shanghai, China
[3]Department of Infectious Diseases, Huashan Hospital Affiliated with Fudan University, Shanghai, China

## AUTHOR ORCIDs

Xinyu Zuo http://orcid.org/0000-0002-4953-288X
Xinyuan Ma https://orcid.org/0009-0006-2858-6957
Richeng Mao http://orcid.org/0000-0001-5534-8299

Jiexian Ma 🆔 http://orcid.org/0000-0002-5102-5158

## AUTHOR CONTRIBUTIONS

Xinyu Zuo, Data curation, Formal analysis, Investigation, Methodology, Resources, Software, Validation, Visualization, Writing – review and editing | Xinyuan Ma, Data curation, Formal analysis, Investigation, Methodology, Resources, Software, Validation, Visualization, Writing – review and editing | Miao Zhang, Investigation, Resources, Software, Writing – review and editing | Richeng Mao, Conceptualization, Data curation, Investigation, Project administration, Resources, Supervision, Validation, Writing – review and editing | Jiexian Ma, Conceptualization, Data curation, Formal analysis, Funding acquisition, Project administration, Software, Supervision, Validation, Writing – original draft, Writing – review and editing

## DATA AVAILABILITY

Materials described in the article, including all relevant raw data, will be freely available to any researcher wishing to use them for non-commercial purposes, without breaching participant confidentiality. Correspondence and requests for materials should be addressed to Jiexian Ma.

## ETHICS APPROVAL

The Huadong Hospital's institutional ethics committee approved the study (2022K095). All procedures involving human subjects complied with the ethical standards of the institutional research committee and the Helsinki declaration, following guidelines approved by the Institutional Review Board (IRB) at Huadong Hospital. All participants provided written informed consent.

## ADDITIONAL FILES

The following material is available online.

### Supplemental Material

**Data S1 (Spectrum02958-24-s0001.doc).** CONSORT 2010 checklist of RCT.
**Data S2 (Spectrum02958-24-s0002.doc).** Trial protocol.
**Supplemental material (Spectrum02958-24-s0003.pdf).** Fig. S1; Table S1.

### Open Peer Review

**PEER REVIEW HISTORY (review-history.pdf).** An accounting of the reviewer comments and feedback.

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
