## [Reviewer comments · Microbiology Spectrum]

Microbiology Spectrum

Novel model to predict risk of invasive fungal infection and fungal prophylaxis timing

Xinyu Zuo, Xinyuan Ma, Miao Zhang, Richeng Mao, and Jiexian Ma

Corresponding Author(s): Jiexian Ma, Huadong Hospital Affiliated to Fudan University

Review Timeline:

Submission Date:	November 18, 2024
Editorial Decision:	February 19, 2025
Revision Received:	April 2, 2025
Editorial Decision:	April 11, 2025
Revision Received:	April 17, 2025
Editorial Decision:	June 12, 2025
Revision Received:	August 1, 2025
Accepted:	August 29, 2025

Editor: Matthew Anderson

Reviewer(s): Disclosure of reviewer identity is with reference to reviewer comments included in decision letter(s). The following individuals involved in review of your submission have agreed to reveal their identity: Shalini Upadhyay (Reviewer #2)

Transaction Report:

DOI: <https://doi.org/10.1128/spectrum.02958-24>

Re: Spectrum02958-24 (Novel clinical model to predict risk of invasive fungal infection and fungal prophylaxis timing determination)

Dear Dr. Jiexian Ma:

Thank you for the privilege of reviewing your work. Below you will find my comments, instructions from the Spectrum editorial office, and the reviewer comments.

Revision Guidelines

Sincerely,
Matthew Anderson
Editor
Microbiology Spectrum

Reviewer #2 (Comments for the Author):

The invasive fungal infections are of significant concern especially for the immunocompromised individuals suffering from co-morbidities. A novel method for predicting the associated risk factors and guiding the prophylaxis can improve the prognosis of such infections.

I found the paper to be overall well written and mostly described in detail. The tables and figures are also well descriptive. Although based on the results of the study the nomogram design seems to be quite useful for the stated purpose, the available literature suggests more risk factors need to be analysed to assess the risk of invasive fungal infections. I suggest the following

revisions:

Major revisions:

1. The title doesn't seem to convey the essence of the paper. In my opinion the title could be modified to sound more connected to the content.
2. The last paragraph of introduction seems repetitive as it has already been covered in the abstract and graphical abstract adequately. The introduction should close with the expression of study objective.
3. Under "Clinical Evaluation and Definition of IFI", "Most patients with probable IFI recovered following antifungal therapy"- this statement doesn't sit well with the results as there are quite a few cases of clinical failures.
4. Under results, subsection- "Incidence of Invasive Fungal Infection was significantly lowered after fungal prophylaxis determined by this model", please explain the statement: "There were 39 patients undergone IFI the non-prophylaxis group and in 9 of the 78 patients (11.5%) in the prophylaxis group."

Minor revisions:

1. "IFI" full form is provided in the first line of Introduction, yet the same thing has been repeated many a times in subsequent text.
2. Full forms of a few abbreviations have not been provided or mentioned after the first use like CAR-T, PI3K, PCR, NGS, AIDS, CNS, HSCT.
3. Under "Clinical Evaluation and Definition of IFI", fifth line, add with- In this study, patients presenting ...with...clinical criteria of IFI.
4. "European Organization for Research and Treatment of Cancer (EORTC)", though it has been abbreviated but throughout the paper the full form has been used.

Reviewer #3 (Comments for the Author):

The manuscript by Zuo et al describes production of a predictive tool to assign patients for fluconazole prophylaxis based on other clinical and demographic characteristics. Overall, the data is presented clearly and all conclusions are well supported. I have a few specific comments and one suggestion to improve the manuscript.

Add line numbering throughout to facilitate the review process.

"nowadays guidelines" is unclear in the Methods section on page 7. Do you mean in current practice?

"The assessment of humoral immune function involved measuring IgG, IgM, and IgA levels using a turbidimetric immunoassay." Is not a complete sentence.

Please include details in the methods on how microscopy and PCR methods for confirmation of IFI was performed.

Inclusion of amphotericin B in either the fluconazole group or placebo group could strongly influence the likelihood of IFI. A specific assessment of this treatment should be assessed between the two groups. If individuals who received AmphoB are removed, does this alter the results?

Table 1 shows no differences between groups for any listed variable. Please edit the line in the Results on page 11 to clarify this point or indicate clinical indicators that are different. Perhaps this could be described as demographics and baseline health condition.

Edit as "There were 39 patients undergoing IFI the non-prophylaxis group and in 9 of the 78 patients (11.5%) in the prophylaxis group."

Recommendations:

Comments to author

Title: Novel clinical model to predict risk of invasive fungal infection and fungal prophylaxis timing determination

The invasive fungal infections are of significant concern especially for the immunocompromised individuals suffering from co-morbidities. A novel method for predicting the associated risk factors and guiding the prophylaxis can improve the prognosis of such infections.

I found the paper to be overall well written and mostly described in detail. The tables and figures are also well descriptive. Although based on the results of the study the nomogram design seems to be quite useful for the stated purpose, the available literature suggests more risk factors need to be analysed to assess the risk of invasive fungal infections. I suggest the following revisions:

Major revisions:

1. The title doesn't seem to convey the essence of the paper. In my opinion the title could be modified to sound more connected to the content.
2. The last paragraph of introduction seems repetitive as it has already been covered in the abstract and graphical abstract adequately. The introduction should close with the expression of study objective.
3. Under "Clinical Evaluation and Definition of IFI", "Most patients with probable IFI recovered following antifungal therapy"- this statement doesn't sit well with the results as there are quite a few cases of clinical failures.
4. Under results, subsection- "Incidence of Invasive Fungal Infection was significantly lowered after fungal prophylaxis determined by this model", please explain the statement: "There were 39 patients undergone IFI the non-prophylaxis group and in 9 of the 78 patients (11.5%) in the prophylaxis group."

Minor revisions:

1. “IFI” full form is provided in the first line of Introduction, yet the same thing has been repeated many a times in subsequent text.
2. Full forms of a few abbreviations have not been provided or mentioned after the first use like CAR-T, PI3K, PCR, NGS, AIDS, CNS, HSCT.
3. Under “Clinical Evaluation and Definition of IFI”, fifth line, add with- In this study, patients presenting ...with...clinical criteria of IFI.
4. “European Organization for Research and Treatment of Cancer (EORTC)”, though it has been abbreviated but throughout the paper the full form has been used.

Response Letter

Dear Prof. Matthew Anderson and reviewers,

Thank you very much for the external review of our manuscript entitled “Novel clinical model to predict risk of invasive fungal infection and fungal prophylaxis timing determination” (Manuscript ID: Spectrum02958-24). We sincerely appreciate the reviewers' constructive comments for this paper, which helps to improve the paper. Based on the instructions provided in your letter, we uploaded the file of the revised manuscript addressed all the concerns. Accordingly, here are our detailed point-by-point response to the reviewers' comments below.

To Reviewer #2:

Major revisions:

1. *The title doesn't seem to convey the essence of the paper. In my opinion the title could be modified to sound more connected to the content.*

Authors' Response: We sincerely appreciate the reviewer's insightful suggestion regarding the improvement of our manuscript title. After careful consideration, we have revised the title from “Novel clinical model to predict risk of invasive fungal infection and fungal prophylaxis timing determination” to “Novel model to predict risk of invasive fungal infection and fungal prophylaxis timing” The refined title better captures our study's dual functionality: 1. Establishing a novel predictive model; 2. Demonstrating its clinical reliability through prospective study validation and conducting fungal prophylaxis according to this model among high risk and low risk group to predict prophylaxis timing. This dual emphasis differentiates it from conventional models limited to retrospective risk assessment.

2. *The last paragraph of introduction seems repetitive as it has already been covered in the abstract and graphical abstract adequately. The introduction should close with the expression of study objective*

Authors' Response: We appreciate the reviewer's precise identification of content overlap between sections. In response, we deleted redundant methodological statements in the introduction's closing paragraph that duplicated content from the abstract and graphical abstract. And we have revised the final paragraph to explicitly foreground the study's primary objectives in Page 5, Line 21.

3. *Under "Clinical Evaluation and Definition of IFI", "Most patients with probable IFI recovered following antifungal therapy"- this statement doesn't sit well with the results as there are quite a few cases of clinical failures.*

Authors' Response: We greatly appreciate the insightful comments provided. The statement regarding patient recovery through antifungal therapy created potential misinterpretation and

compromised conceptual coherence within the methodological framework. Based on the aforementioned rationale, we have removed this content to preserve contextual consistency.

4. *Under results, subsection- "Incidence of Invasive Fungal Infection was significantly lowered after fungal prophylaxis determined by this model", please explain the statement: "There were 39 patients undergone IFI the non-prophylaxis group and in 9 of the 78 patients (11.5%) in the prophylaxis group."*

Authors' Response: Thank you for your constructive feedback. We have revised the sentence to address the grammatical concern while ensuring scientific accuracy. The observed reduction in IFI incidence provides robust evidence for the efficacy of antifungal prophylaxis, particularly in delivering clinically significant protection to immunocompromised patient cohorts. These findings are elaborated in the revised manuscript in Page 14 Line 31-Page 15 Line 5.

Minor revisions:

1. *"IFI" full form is provided in the first line of Introduction, yet the same thing has been repeated many a times in subsequent text.*

Authors' Response: We sincerely appreciate your meticulous review of terminology standardization. We have standardized the use of "IFI" throughout the manuscript. The full term "invasive fungal infection" is now retained exclusively in the opening paragraph of the Introduction (Page 4, Line 2), with strict adherence to abbreviation conventions. All redundant expansions in subsequent sections have been removed.

2. *Full forms of a few abbreviations have not been provided or mentioned after the first use like CAR-T, PI3K, PCR, NGS, AIDS, CNS, HSCT*

Authors' Response: Thank you for your thorough review and valuable suggestions regarding abbreviation standardization in our manuscript. We have carefully revised all abbreviations according to the guideline stating that "abbreviations should be defined only when terms recur ≥ 3 times; terms mentioned ≤ 2 times retain their full spelling to prevent reader confusion." And all abbreviations now receive explicit definitions at first mention (e.g., "next-generation sequencing (NGS)" and "bronchoalveolar lavage fluid (BALF)").

3. *Under "Clinical Evaluation and Definition of IFI", fifth line, add with- In this study, patients presenting ...with...clinical criteria of IFI.*

Authors' Response: Thank you for your thoughtful and constructive comments. We have revised the sentence structures to enhance clarity and flow in Page 8 Line 5-8.

4. *"European Organization for Research and Treatment of Cancer (EORTC)", though it has been abbreviated but throughout the paper the full form has been used.*

Authors' Response: Thank you for your feedback. We have revised the manuscript to ensure

consistent use of the abbreviation "EORTC" after its first definition. All subsequent mentions of the full term have been replaced with "EORTC" to avoid redundancy.

To Reviewer #3:

1. *"Nowadays guidelines" is unclear in the Methods section on page 7. Do you mean in current practice?*

Authors' Response: Thank you for highlighting this ambiguity. We have revised the phrase "Nowadays guidelines" to "current clinical practice guidelines" in Page 7, Line 8-10 of the revised manuscript. This modification clarifies that we are referring to contemporary, evidence-based guidelines widely adopted in the field, aligning with your suggestion.

2. *"The assessment of humoral immune function involved measuring IgG, IgM, and IgA levels using a turbidimetric immunoassay." Is not a complete sentence*

Authors' Response: Thank you sincerely for your attentive feedback. We deeply appreciate your suggestion to improve the clarity of this sentence. We have revised the sentence to "To evaluate humoral immune function, we involved measuring IgG, IgM, and IgA levels using a turbidimetric immunoassay." In Page 7, Line 10-12.

3. *Please include details in the methods on how microscopy and PCR methods for confirmation of IFI was performed.*

Authors' Response: Thank you for this excellent suggestion, which has significantly enriched our work. We have expanded the "Clinical Evaluation and Definition of IFI" subsection in the Methods (Page 8, Lines 5-22 of the revised manuscript) to include direct microscopy and real-time PCR. Your feedback improved the methodological clarity significantly.

4. *Inclusion of amphotericin B in either the fluconazole group or placebo group could strongly influence the likelihood of IFI. A specific assessment of this treatment should be assessed between the two groups. If individuals who received AmphoB are removed, does this alter the results?*

Authors' Response: We appreciate your insightful comments regarding the use of amphotericin B in our study. We would like to clarify the role of amphotericin B in our study design.

Current guidelines from the Infectious Diseases Society of America (IDSA), European Society of Clinical Microbiology and Infectious Diseases (ESCMID), European Confederation of Medical Mycology (ECMM), and European Respiratory Society (ERS) recommend amphotericin B as a first-line agent for empirical and diagnostic-driven treatment of invasive candidiasis and aspergillosis. Moreover, due to the severe toxic side effects of amphotericin B, we did not select it for fungal prophylaxis in our study. For patients who developed fungal infections in both the prophylaxis and placebo groups, we typically followed clinical guidelines to determine

appropriate therapeutic agents based on the specific type of fungal infection. Amphotericin B liposomal was only considered in cases of poor response to conventional antifungal agents or specific infections such as *Rhizopus*. Consequently, the usage of this medication was minimal in both cohorts, and its limited application is unlikely to have significantly influenced the study outcomes. Therefore, we did not conduct further analysis regarding its potential impact.

5. *Table 1 shows no differences between groups for any listed variable. Please edit the line in the Results on page 11 to clarify this point or indicate clinical indicators that are different. Perhaps this could be described as demographics and baseline health condition.*

Authors' Response: Thank you for your valuable suggestion. We sincerely apologize for any confusion caused by the redundant phrasing in our original manuscript. And we confirm no statistically significant differences in baseline variables between the training cohort and validation cohort. To enhance readability while preserving scientific accuracy, we streamlined the text and revised terminology to “demographics and baseline health condition” as recommended in Page 12 Line 7-8.

6. *Edit as "There were 39 patients undergoing IFI the non-prophylaxis group and in 9 of the 78 patients (11.5%) in the prophylaxis group."*

Authors' Response: Thank you for your meticulous review. We have expanded the original sentence to improve readability while ensuring scientific accuracy in Page 14 Line 30-Page 15 Line 3.

We would like to thank you once again for your thoughtful and constructive comments. We believe the revisions have improved the manuscript significantly and we are confident that it is now more aligned with the journal's standards. We appreciate your time and effort in reviewing our work and look forward to your further feedback.

Please do not hesitate to contact us if any additional information or clarification is needed.

Sincerely,

Jiexian Ma.

majiexian@fudan.edu.cn

Re: Spectrum02958-24R1 (Novel model to predict risk of invasive fungal infection and fungal prophylaxis timing)

Dear Dr. Jiexian Ma:

Thank you for the privilege of reviewing your work. Below you will find my comments, instructions from the Spectrum editorial office, and the reviewer comments.

Revision Guidelines

Sincerely,
Matthew Anderson
Editor
Microbiology Spectrum

Reviewer #2 (Comments for the Author):

All the major and minor revisions suggested by me have been updated in the manuscript. I am fully satisfied with the revisions.

Reviewer #3 (Comments for the Author):

Thank you for providing the response to the comments provided in the initial review. Some concerns remain.

1. The fact that amphotericin B was not a core component of the study design does not negate its potential influence on the study. Its potential impact on patient outcome must be accounted for in the analysis. It confounds the conclusions drawn by the authors.
2. The description of the patients (prior point 6) has been clarified but the number of individuals in each of the control and treatment groups do not add up appropriately. This sections requires revision.
3. Please revise the manuscript for grammar. There are many grammatical mistakes throughout. I would advocate for an English-language review of the manuscript once the changes are incorporated.

Response Letter

Dear Prof. Matthew Anderson and reviewers,

Thank you very much for the external review of our manuscript entitled “Novel model to predict risk of invasive fungal infection and fungal prophylaxis timing” (Manuscript ID: Spectrum02958-24R1). We appreciate the opportunity to revise our manuscript and thank you for your thoughtful comments and suggestions. In this revised version, we have carefully addressed each point raised, and we believe that the changes have significantly strengthened the manuscript. Below, we provide a detailed response to each reviewer comment.

To Reviewer #2:

1. *All the major and minor revisions suggested by me have been updated in the manuscript. I am fully satisfied with the revisions.*

Authors' Response: Thank you very much for your thorough review and valuable comments. We appreciate your positive feedback, and we are pleased to hear that you are fully satisfied with the revisions made in response to your suggestions. Your insightful recommendations have greatly helped us to improve the manuscript, and we are grateful for the time and effort you have invested in reviewing our work.

To Reviewer #3:

1. *The fact that amphotericin B was not a core component of the study design does not negate its potential influence on the study. Its potential impact on patient outcome must be accounted for in the analysis. It confounds the conclusions drawn by the authors.*

Authors' Response: Thank you for pointing out the potential influence of amphotericin B on our study outcomes. We would like to clarify that amphotericin B was administered only in the treatment phase for 2 out of the 163 patients in our cohort, and these patients were included in either the fluconazole or placebo group according to the randomization protocol. Therefore, amphotericin B was not part of the preventive strategy under investigation.

Given that only 2 patients received amphotericin B and that its use was limited to the treatment of infections, we believe its potential impact on the overall study conclusions is minimal. Additionally, we conducted supplementary analyses excluding these two cases, and the results remained consistent, further supporting the robustness of our findings.

While amphotericin B was not a predefined variable in our study design, we acknowledge the importance of considering such factors, and we will take this into account in future analyses and discussions.

2. *The description of the patients (prior point 6) has been clarified but the number of individuals in each of the control and treatment groups do not add up appropriately. This section requires*

revision.

Authors' Response: Thank you for your valuable comment regarding the discrepancy in participant numbers between the control and treatment groups. In the Fungal Prophylaxis group, a total of 83 patients were enrolled. However, 4 patients were excluded from the analysis of invasive fungal infection (IFI) incidence due to adverse effects of long-term fluconazole administration, such as liver dysfunction, and 1 patient withdrawal for a strong desire to discontinue the trial. As a result, the IFI incidence analysis included 78 patients. We apologize for any confusion our initial explanation may have caused. Thank you for pointing out the inconsistency in the group numbers.

Your comment has helped us realize that, although these 5 patients did not complete the study, they did receive fungal prophylaxis and should still be acknowledged as part of the prophylaxis group. To avoid confusion for readers, we have revised the relevant section of the manuscript to clarify this distinction more explicitly in Page 12 Line 28.

3. *Please revise the manuscript for grammar. There are many grammatical mistakes throughout. I would advocate for an English-language review of the manuscript once the changes are incorporated.*

Authors' Response: Thank you for your valuable feedback regarding the grammatical issues in our manuscript. We apologize for any confusion our submission may have caused. To address this, we have thoroughly revised the manuscript to correct grammatical errors and improve overall clarity. Additionally, we sought assistance from a professional English-language professor to review and enhance the language quality of our manuscript. We believe these revisions have significantly improved the readability and quality of our work.

We sincerely thank you for your time and constructive feedback. We have addressed all the comments and suggestions provided, and we believe the manuscript has been significantly improved.

We look forward to hearing from you regarding our revised submission.

Sincerely,

Jiexian Ma.

majiexian@fudan.edu.cn

Re: Spectrum02958-24R2 (Novel model to predict risk of invasive fungal infection and fungal prophylaxis timing)

Dear Dr. Jiexian Ma:

Thank you for the privilege of reviewing your work. Below you will find my comments, instructions from the Spectrum editorial office, and the reviewer comments.

Revision Guidelines

Sincerely,
Matthew Anderson
Editor
Microbiology Spectrum

Reviewer #3 (Comments for the Author):

I thank the authors for addressing point #2 and #3 in full.

Data supporting point 1 was not included in the Response to Reviewer to demonstrate the lack of impact of amphiB. It was stated that it had no effect after the analysis. without the data to back this up, the statement remains unresolved. Please provide the data to support this statement.

Also, Supplemental Figure 1 has Chinese characters on the x-axis. This should be fixed.

Response Letter

Dear Prof. Matthew Anderson and reviewers,

We are truly grateful for the external review of our manuscript entitled “Novel model to predict risk of invasive fungal infection and fungal prophylaxis timing” (Manuscript ID: Spectrum02958-24R2). We sincerely appreciate the reviewers’ insightful comments and constructive suggestions. In this revised version, we have thoroughly addressed all the points raised, and we believe the revisions have substantially improved the quality of the manuscript. A detailed point-by-point response to each comment is provided below.

To Reviewer #3:

- Data supporting point 1 was not included in the Response to Reviewer to demonstrate the lack of impact of amphiB. It was stated that it had no effect after the analysis. without the data to back this up, the statement remains unresolved. Please provide the data to support this statement.*

Authors' Response: We sincerely thank the reviewer for their thoughtful re-evaluation of our manuscript. We apologize for not including the supporting data in our previous response regarding the impact of amphotericin B on patient outcomes.

The relevant tables have been included in the response letter. In the following analysis, we excluded patients who received amphotericin B. The data demonstrate that the administration of amphotericin B had no significant effect on clinical outcomes in patients with suspected IFI. These results support our previous conclusion that amphotericin B use did not influence the overall prognosis in our study population. The treatment regimens are presented in the appendix, confirming that there was no bias in the administration of antifungal therapy.

We appreciate the reviewer’s valuable comment, which helped us clarify this point with supporting evidence.

Table 1 Clinical characteristics in high risk IFI patients with fluconazole prophylaxis and non-prophylaxis

	High-risk Patients (N=161)		p value
	Non-fungal prophylaxis (N=78)	Fungal prophylaxis (N=83)	
Age (mean, range)	69.91(31-88)	66.58 (21-87)	0.056
Male (Yes/Total)	56(71.8%)	47 (56.6%)	0.133
Central Venous Catheter (Yes/Total)	25(32.1%)	16 (19.3%)	0.286
Transplantation (Yes/Total)	4(5.1%)	11 (13.3%)	0.061
Diabetes (Yes/Total)	16(20.5)	16 (19.3%)	0.936
IgG (mean, range) (g/L)	11.12(3.13-24.7)	10.48 (3.46-71.05)	0.599
CD4 (mean, range) (µl)	295.54(15.00-496.00)	337.24 (20.93-5114.38)	0.515

WBC (mean, range) ($\times 10^9/L$)	7.41(1.64-27.38)	6.59 (1.20-70.80)	0.227
Neutrophil (mean, range)	6.81(0.68-52.97)	4.55 (0.51-62.05)	0.062
Lymphocyte (mean, range)	0.88(0.12-4.85)	0.96 (0.17-2.52)	0.415

Table 2 Clinical Response and Reasons for Failure during the Treatment Phase.

Clinical Response	Non-fungal Prophylaxis	Fungal Prophylaxis	P Value
	Patients (Placebo)	Patients	
	(N=78)	(N=83)	
Clinical success	41	67	
Clinical failure	37	14	0.003
Proven or probable invasive fungal infection	37	9	<0.001
Adverse event possibly or probably related to study treatment, resulting in discontinuation	0	4	0.055
Withdrawal for any reason and loss to follow-up	0	1	0.33

Table 3 Proven or Probable Invasive Fungal Infection during the Treatment Phase

Species	Low-risk Patients (N=102)	High-risk Patients (N=161)		p value
		Non-fungal Prophylaxis (N=78)	Fungal Prophylaxis (N=83)	
Mold	7	8	5	P=0.14
Aspergillus fumigatus	5	5	1	
A.flavus	0	2	1	
A.terreus	0	1	1	
A.niger	1	0	0	
A.tubingensis	1	0	0	
Yeast	5	26	3	P<0.01
Candida albicans	3	12	1	
C.glabrata	0	1	0	
C.parapsilosis	0	5	1	
C.tropicalis	2	6	1	
C.krusei	0	2	0	
Other	0	3	1	P=0.10

Pneumocystis jirovecii	0	3	1
Scedosporium	0	0	0

2. *Supplemental Figure 1 has Chinese characters on the x-axis. This should be fixed.*

Authors' Response: We thank the reviewer for the careful review and helpful suggestion. We have corrected the x-axis labels in Supplementary Figure 1, and all Chinese characters have been replaced with the appropriate English terms. This revision ensures clarity and consistency for all readers.

We look forward to hearing from you regarding our revised submission.

Sincerely,

Jiexian Ma.

majiexian@fudan.edu.cn

Re: Spectrum02958-24R3 (Novel model to predict risk of invasive fungal infection and fungal prophylaxis timing)

Dear Dr. Jiexian Ma:

Your manuscript has been accepted, and I am forwarding it to the ASM production staff for publication. Your paper will first be checked to make sure all elements meet the technical requirements. ASM staff will contact you if anything needs to be revised before copyediting and production can begin. Otherwise, you will be notified when your proofs are ready to be viewed.

Sincerely,
Matthew Anderson
Editor
Microbiology Spectrum